# Global Environmental Health Impacts of Rare Earth Metals: Insights for Research and Policy Making in Africa

Habeebullah Jayeola Oladipo [1,2,3,4,*], Yusuf Amuda Tajudeen [1,3,4,5,*], Emmanuel O. Taiwo [6], Abdulbasit Opeyemi Muili [7], Rashidat Onyinoyi Yusuf [2], Sarat Ayomide Jimoh [2], Muhammad Kamaldeen Oladipo [2], Iyiola Olatunji Oladunjoye [1,8], Oluwaseyi Muyiwa Egbewande [2], Yusuff Inaolaji Sodiq [9], Abdulhakeem Funsho Ahmed [10,11] and Mona Said El-Sherbini [4,12,13,*]

1    Department of Microbiology, Faculty of Life Sciences, University of Ilorin, P.M.B. 1515, Ilorin 240003, Nigeria
2    Faculty of Pharmaceutical Sciences, University of Ilorin, P.M.B. 1515, Ilorin 240003, Nigeria
3    Planetary Health Alliance Campus, Boston, MA 02115, USA
4    Nova Network, Baltimore, MD 21231, USA
5    Department of Epidemiology and Medical Statistics, Faculty of Public Health, College of Medicine, University of Ibadan, P.M.B. 5017, Ibadan 200212, Nigeria
6    Foreign, Commonwealth and Development Office, Maitama, Abuja 900103, Nigeria
7    Department of Medicine, Ladoke Akintola University of Technology, Oyo/Ilorin Rd, Ogbomosho 210214, Nigeria
8    Rouleaux Foundation, Agege 100283, Nigeria
9    Department of Medicine, Faculty of Clinical Sciences, Obafemi Awolowo University, Ibadan-Ife Rd, Ife 220101, Nigeria
10   Institute of Basic and Applied Science, Department of Science Laboratory Technology, Kwara State Polytechnic, P.M.B. 1375, Ilorin 241103, Nigeria
11   Faculty of Health Sciences, Department of Public Health, Al-Hikmah University, Ilorin 240281, Nigeria
12   Department of Medical Parasitology, Faculty of Medicine, Cairo University, Cairo 11562, Egypt
13   Invited Faculty, the Nova Institute for Health, Baltimore, MD 21231, USA
*    Correspondence: oladipohabeebullah@gmail.com (H.J.O.); tajudeenamudayusuf@gmail.com (Y.A.T.); monas.elsherbini@kasralainy.edu.eg (M.S.E.-S.); Tel.: +234-(0)-8179122773 (H.J.O.); +234-(0)-706-206-3691 (Y.A.T.); +20-100-246-5704 (M.S.E.-S.)

**Abstract:** The rise of globalization and industrialization has driven the demand for rare earth metals (REMs). These metals are widely used in various sectors of the global economy with various applications in medicine, renewable energy, electronics, agriculture, and the military. REMs are likely to remain an important part of our global future, and, as production increases, areas contaminated by REMs are expected to expand over the coming decades. Thus, triggering significant adverse environmental, animal, and human health impacts. Despite increased attention on REMs outside China in recent years, there are limited studies exploring REM production, deposits, and associated health impacts in the African context. Proper mine management, adequate safety protocols, sustainable processing methods, and waste handling systems have been identified and proposed globally; however, the nature and scale of implementing these management protocols on the African continent have been less clear. Therefore, planetary health-centered solutions are urgently needed to be undertaken by researchers, policy makers, and non-governmental actors in Africa and across the globe. This is with the overarching aim of ensuring eco-friendly alternatives and public health consciousness on REM exploitations and hazards for future generations to come.

**Keywords:** rare earth metals; environmental health impacts; policy and research; planetary health; Africa

## 1. Introduction

The rise of global urbanization and industrialization has driven the demand for rare earth metals (REMs), also known as rare earth elements (REEs). These elements are widely used in various sectors of the global economy, including medicine, science and technology,

energy development, and agriculture, and few studies focus on the consequences of long-term occupational exposure to REMs on human health [1]. Generally, REMs are a group of eighteen metallic elements that comprise fifteen elements in the lanthanide series, in addition to scandium, indium, and yttrium. Rare earth metals have had a significant impact on medicine, technology, agriculture, and infrastructure, and they are utilized in a variety of products. These metallic elements have been used variously in the production of electronics, hybrid cars, renewable energy systems, and other technologies [1,2]. Although they are found in group 17 of the periodic table, they are not particularly rare compared to other elements, but they are much less abundant than other metals such as iron or copper and are also difficult to extract [2]. Based on current understanding and research in the literature, there is no universally accepted formal definition of REM, but most of the literature defines them as any member of the lanthanide series that also exhibits high-performance rates [2]. There are various REM deposits, which include carbonatite, alkaline and per-alkaline igneous rocks, iron oxide copper-gold deposits, veins, and skam deposits, among others [3]. Carbonatite is the primary source of REMs worldwide, followed by alkaline igneous rocks, Iron Oxide Copper Gold (IOCG) ore deposits, placers, and ion adsorption clay deposits [4]. The major REM deposits in Africa include carbonatite, alkaline igneous rock-associated, alluvial placer, hydrothermal deposit, and marine placer. These deposits are distributed across the twenty-one REM reservoirs in Africa. South Africa has the highest, and Namibia comes in second position with five deposits [5]. Other nations with prospective REM repositories include Morocco, Mauritania, Egypt, Burundi, Kenya, Tanzania, Malawi, Mozambique, and Madagascar [6]. Due to the inadequate human and infrastructure resources for the discovery and mining of REMs, most of these facilities are now unproductive. Most REM resources in Africa are found within Archean cratons and alkaline igneous provinces [6].

Human and animal exposures to these elements have increased in the last decade through food and water contamination, given their widespread use in a range of industrial and consumer products. The potential toxicity of REMs has been increasingly reported to be a health hazard if ingested or absorbed by the body through environmental contamination. For example, chronic interstitial lung disease and pneumoconiosis have been documented and reported in REM miners as occupational hazards [1,2]. In this article, we review what is known about the bioavailability and toxicity of REMs at the interface between human, animal, and environmental health. This expands to include a range of detrimental health effects, such as immunotoxicity, carcinogenicity, and neurotoxicity [2]. The geographic expanse of global REM research was also addressed, highlighting countries and regions that feature prominently in the literature on REMs. Finally, research coverage in Africa, future research, and policy-making on the resource-rich continent was discussed through the planetary health lens.

## 2. Materials and Methods

In this narrative review, we conducted a Web-based search for peer-reviewed journal articles published in English from 1999 to 2023. In our search, relevant databases and free-web article searches from PubMed (https://pubmed.ncbi.nlm.nih.gov/; accessed on 2 January 2023), ScienceDirect (https://www.sciencedirect.com; accessed on 4 February 2023), Wiley (https://www.wiley.com/en-us; accessed on 6 February 2023), and Google Scholar (https://scholar.google.com; accessed on 12 February 2023) were considered for suitable articles. The following keywords were used during the web and databases searches: "Rare Earth Elements", "Rare Earth Metals", "Environmental Health Impacts", "Human and Animal Health Impacts", "Policy and Research", "Land-use Changes", "Planetary Health", and "Africa". Papers were selected by reviewing their abstracts and title and using supplementary references obtained from the list of references on the papers as an additional resource. The inclusion criteria set for this review were as follows; Restricted to broad-context papers on Rare Earth Metals; Policy Making; Planetary Health; Africa; and related studies that involved the impact of Rare Earth Metals on Environmental, Human,

and Animal health were considered. Papers that did not meet any of the criteria above were excluded, and the supplementary references used in those papers that met any of the criteria above were searched for and reviewed appropriately. The list of references from relevant articles was checked for additional articles used in the review. A cumulative number of 66 articles was considered for the final review. All articles included in the final review established incontrovertible proof that the exploitation and production of rare earth metals are intensifying globally and are important factors that trigger poor environmental health safeguards in Africa and global health at large.

### 3. Background History of Rare Earth Metals

The history of REMs began in the late 18th century when Swedish artillery officer Karl Axel Arrhenius found a dark mineral in a feldspar pit near Ytterby—a village in Sweden—close to Stockholm, in 1787 [3]. It was called "rare" because it had never been seen before, and "earth" because it was the term used to name rocks in the 18th century. Gadolin, in 1794, named the unknown dark mineral 'earth yttria', after the town where it was discovered [4]. In the 19th century when the discovery and naming of elements were popular, Swedish ironmaster and scientist—Wilhelm Hisinger together with a young chemist—Jons Jakob Berzelius, in the year 1803, analyzed a sample of tungsten (heavy stone) of bastanas and isolated ceritic earth similar to yttria; the Swedish chemist named this new element ceria from the asteroid ceres, discovered two years before [7].

Quite sometime after Hisinger and Berzelius discovered ceritic earth, a Swedish chemist, Carl Gustav Mosander (1797–1858), in 1839, separated ceritic earth into three different parts, which include cerium (ceroxide), lanthanum (lanthanumoxide), and didym (didym-earth). The name for the element lanthanum comes from the Greek 'lanthanein', which means 'being hidden'. In addition, the name didym originated from the Greek word 'didymos', that is, twin, and was chosen because of the similarity of didym with lanthanum [3].

In the second half of the nineteenth century, the chemists Gustav Kirchoff and Robert Bunsen developed spectroscopy as a technique to identify elements using the phenomenon of light spectroscopy [4], since the major challenge after Mosander's work was finding ways to separate the elements.

In 1880 while a student of Robert Bunsen, Carl Auer Von Welsbach showed that didymium, which was then thought to be an element, was, in fact, an alloy of two REEs which he named neodymium and praseodymium, and as he shifted his interest into industrial concerns, he became the pioneer in developing a commercial use of REEs [4]. His technological and commercial success sparked greater interest in the broader applications of REEs, which expanded the rare earth industry dramatically and drove the quest for raw materials beyond Europe, to the Americas, colonial India, and China [8].

In the 20th century, REMs took on a new scientific and geopolitical importance to the advancement in atomic physics [4]. This is due to the complexity of the spectra of the rare earth and the questionable purity of the samples. Many claims were being made for new elements that proved false; however, despite this challenge, by 1901, only two rare earth elements had not yet been discovered. In 1907, Georges Urbain, Carl Auer Von Welsbach, and Charles James separated ytterbium into two fractions; Urbain named the new element lutetium, which was the 16th REE to be discovered [9]. The isolation and identification of the final REM were finally made in 1947 by Jacob Akiba Marinsky, Lawrence Elgin Glendelin, and Charles Dubois Coryrll from the nuclear fusion of uranium; the element was subsequently named promethium after Prometheus—the son of Iapetus, because it was one of the first atomic fires created by man [8]. From the 1950s up to the 2000s, learning how to use REMs was the main quest due to their importance in the production of powerful magnets, lasers, red phosphorus, nickel hydride batteries, and portable electronics [4]. REMs are likely to remain an important part of our future, from quantum computing and material science to medical applications and green technology [4], not least in the

production of much-needed alternative energy technologies as the world moves toward a net zero, decarbonized future.

## 4. Analysis and Exploration of Rare Earth Metals

Rare earth metals (REMs) or rare earth elements (REEs) are known as an abundant group of eighteen elements that comprise fifteen elements in the lanthanide series, in addition to scandium, indium, and yttrium. Some REMs are relatively abundant in the Earth's crust, but not often at concentrations high enough to make mining currently economically viable [8]. REEs are further categorized into two sub- groups: light rare earth elements (LREEs) and heavy rare earth elements (HREEs). Elements from Lanthanum (57) to Europium (63) are known as LREEs, while the elements from Gadolinium (64) to Lutetium (71) including Yttrium are known as HREE. These two categories of REEs occur in the same deposits altogether except Scandium (21), which is categorized as a transition metal. Furthermore, LREEs are more abundant than HREEs [10,11].

The major terrestrial sources of these elements are found in the mineral deposits of Bastnaesite and monazite. Bastnaesite deposits in the United States and China are the highest concentrations of REEs, while monazite deposits in Australia, South Africa, China, Brazil, Malaysia, and India constitute the second highest concentration of this resource [12].

Vast exploration of REMs began in the 1950s with extensive mining carried out in Brazil and India [10]. Subsequently, it was South Africa that pushed its way to the forefront of the REM mining industry but was soon overtaken by the Mountain Pass mine in the United States of America, more precisely in the state of California. The United States dominated the world of exploration until the late 20th century [12], when Xu Guangxian's work on isolating uranium led to the discovery of the cascade theory of countercurrent extraction, which revolutionized the production of REM and sparked a boom in the mining of REMs in China in the 1970s [13].

In the last decade, there has been a major rise and fall in the exploration of REMs. In the year 2010 to 2014, there was a global REMs boom that yielded outstanding results in terms of newly developed REM resources in other countries but was short-lived due to the various risks in exploration and processing [14]. Major risks due to the issues associated with the mining and processing of REMs are the environmental hazards caused by radioactive waste from thorium present in the ores, which tend to contaminate water and air. Currently, China has the largest share of the production and processing of REMs, having largely ignored the effects of the associated environmental impacts over the years [15].

## 5. Applications of Rare Earth Metals

REMs have found various applications in medicine, renewable energy technology, agriculture, and the military. In medicine, the light-emitting properties of REMs make them suitable for the development of luminescent materials which are used in scintillators for medical imaging [16]. Materials containing lanthanide ions as activators give rise to strong light ray emission in the spectral region from ultraviolet ray to infrared ray upon excitation with ionizing radiation that is, by itself, very difficult to detect. Therefore, lanthanum-activated scintillators can efficiently convert X-rays or gamma rays to light rays that are easily detected by photodetectors [16]. Other unique properties of REMs, such as radiation emission or magnetism, allow REMs to be used in many different therapeutic and diagnostic applications in modern medicine [17].

In agriculture, REMs are used as fertilizers to improve crop growth and yield and, therefore, lead to an additional increase in REM concentrations in the soil [18]. However, several studies have shown the beneficial effects of REMs at low concentrations and harmful effects at higher levels in terrestrial plants and in the water among aquatic algae and plants [19].

In the military, REMs have been found in various usages in permanent magnets, REMs are found in two permanent magnet materials: samarium cobalt and neodymium iron boron. Magnets made from the REM (Neodymium) are considered the world's

strongest permanent magnets and are essential to many military weapons systems, while the Samarium Cobalt magnet has a high magnetic retention property which makes it retain its magnetism at high temperatures; these are useful in various military weapons such as precision-guided missiles, smart bombs, and aircraft. Additionally, the high strength of the neodymium iron boron magnet makes it useful in making light magnets in the military weapon system [20].

REMs are also used extensively in renewable energy and contemporary consumer electronic devices. With improvements in technology and increasing demand for cleaner, greener, and more efficient energy sources, REMs have been used in various renewable energies such as solar and wind energy. REMs such as neodymium, terbium, indium, dysprosium, and praseodymium are used as components of photovoltaic cells, used in the generation of solar electricity [21]. Additionally, Neodymium and Dysprosium are permanent magnets employed in synchronous generators in direct-drive wind turbines [22]. These magnets allow for more lightweight designs and compact direct-drive wind turbines compared to gearbox wind turbines [23]. Furthermore, REMs are used as components in technological devices such as smartphones, magnets, computers, fluorescent and light-emitting diodes, and electronic displays [24].

## 6. Environmental Challenges Associated with Mining Rare Earth Metals

Rare earth metals (REMs) are commonly found in a variety of accessory minerals, such as phosphates, carbonates, fluorides, and silicates. They do not occur alone naturally, but only as a part of the host mineral's chemistry. As a result, REMs are usually mined through complex processing methods to chemically break down the minerals containing the REMs [25]. Another option for REM extraction is from alternative sources, such as industrial waste and recycling end-of-life REM-bearing products, or replacing less common REM [25]. The principal flow sheets of REM production include five steps: (1) mining of mineral ore bearing REM, including silicates, fluorocarbonates, oxides/hydrogen oxides, and phosphates; (2) beneficiation of the ore mixtures; (3) separation of REM ores into individual REM minerals; (4) reduction in REM minerals into individual REM; and (5) refinement of individual REM to purify REMs [26–28].

REM production involves high energy and resource consumption, high levels of pollution, and significant environmental impacts, and processing is also much more complex than the production of other metals [26,28]. For example, the extraction of 1 ton (t) of REE requires 4.41 t of sulfuric acid, 12.32 t of sodium chloride, 1.64 t of sodium hydroxide, 1.17 t of hydrochloric acid, 1.90 t of water, and the grinding of almost 50 t of mineral ore in Bayan Obo in Mongolia, and generally, the energy used to obtain 1 t of an individual REE varies from 38 to 48 GJ, except for scandium (148 $GJ \cdot t^{-1}$) and yttrium (75 $GJ \cdot t^{-1}$) [29]. Though these environmental issues mentioned above may be produced by other mining activities, the radioactive pollution and toxicity associated with REMs production are specific environmental risks that are attracting more attention globally [30,31].

## 7. Impact of Rare Earth Metals on Global Health

Rare earth metals are likely to remain an important part of our future, and as REM exploration, production, and utilization increases, it is anticipated that the areas contaminated by REM deposits will continue to expand over the coming decades, thus, leading to a significant impact on environmental, animal, and human health.

### 7.1. Impacts on Environmental Health

Mining of REMs is a complex process that can have significant environmental and health impacts. However, in most developing countries, the environmental consequences of mining are often not given enough consideration [32]. Some of the environmental impacts of extracting REMs from ore include deforestation, soil erosion, water contamination, destruction of wildlife habitats, and changes in landscape structure [32]. The type of rock surrounding the REM deposit, the presence of other metals or agents within the rocks, the

climate conditions, and the proximity of streams and lakes to the mine all affect the impacts of REMs [33]. Mining of REMs also results in the destruction of vegetation in mining areas because the share of agricultural land and water quality deteriorates due to the expansion of the mining sites [34,35].

Bioaccumulation, radiation exposure, species invasion, and biodiversity loss are also some of the effects of REMs on environmental health [34–37]. The accumulation of REMs in soils and their use in agriculture as microelement fertilizers have caused toxic effects on the soil microfauna community [25]. This leads to bioaccumulation in crops and the entire food chain, resulting in more and more REMs in ecosystems [25]. In addition, the active migration of pollutants leads to landscape degradation, deterioration of the quality of agricultural produce, soils, water bodies, and the quality of life of individuals and communities living in proximity to mining enterprises [38]. The REMs mining process can also generate large amounts of waste rock and tailings, which can contain radioactive materials and heavy metals. These materials can contaminate the soil, water, and air, posing a risk to human health and the environment. The processing of REMs requires the use of chemicals and acids, which can be toxic and hazardous if not handled properly. The disposal of used chemicals and waste from REM processing can create hazardous waste sites that contaminate soil and groundwater [32]. Moreover, the use of REMs in technology can also have indirect environmental and health impacts through e-waste, which contains rare earth metals and other toxic materials. Improper disposal of e-waste can lead to environmental contamination and health hazards for nearby communities [33]. Therefore, developing sustainable practices for the extraction and use of REMs is crucial to minimize their environmental and health effects. Consumers can also help reduce the impact of REMs by properly disposing of electronic devices and supporting companies that prioritize sustainability and responsible manufacturing.

*7.2. Impacts on Human Health*

REMs are essential components of many high-tech devices and modern technology. However, the mining, processing, and disposal of rare earth metals can have significant negative impacts on human health [34]. Mining activities, such as cutting, drilling, blasting, transportation, processing, and stockpiles, involved in REM production can release dust, small particulate materials, harmful pollutants, and toxic chemicals into the environment. These pollutants can contaminate soil, water, and air, leading to health problems for people living in or near mining areas [34]. Substances released into the air are usually due to the removal of overburden (the removal of rocks and soil that cover the surface to access ores) and rock piles [39]. These substances normally do not contain high concentrations of metals and other minerals but carry this risk of causing chronic lung disease (such as pneumoconiosis) and lung cancer to those exposed to these substances [40]. For example, crystal silica; fine particulate dust was associated with lung cancer and irreversible interstitial pulmonary disease as an occupational hazard in miners or communities in proximity to mining fields [41].

Ores that contain REMs may also contain radioactive material, which can be liberated into the air during mining activities [17]. Miners are susceptible to inhaling, absorbing, and ingesting chemicals throughout the mining process, as they are generally faced with high levels of exposure to REMs [41,42]. For example, many cases of lead poisoning were reported among children, artisanal and small-scale miners, and their family members in Zamfara, Nigeria, in 2010, tragically resulting in the loss of many lives [43]. At these high levels of exposure, vulnerable populations are at increased risk of cardiovascular, pulmonary, neurological, hepatic, and kidney diseases. This also includes leukemia, abnormal levels of some immune cells in complete blood count, chronic gastrointestinal disorders, keratosis, and skin cancers that were also reported in long-term REM exposure studies [38,44–47]. Improper disposal of e-waste can also lead to contaminated soil and water, with serious health problems [42]. Therefore, it is essential to ensure that mining

and processing activities are carried out safely and sustainably and that electronic waste is properly disposed to minimize the negative health consequences.

*7.3. Impacts on Animal Health*

REMs have been used as feed additives in farm animals due to their effectiveness in increasing body weight, feed conversion rate, and milk and egg production in cattle and poultry [48]. However, the long-term effects of this small but prolonged intake of metals on animal health have been a growing concern.

In the beginning, the administration of REMs to farm animals as a growth-promoting factor manifested itself as increased milk production in cows and egg production in laying birds [49]. However, over time, when REMs have accumulated substantially inside the bodies of these animals, it was discovered that the animals suffered from slow health deterioration, including, but not limited to, loss of motor function and epileptic fits, necrosis in liver tissues, hyperactive lymphoid follicles, damage to the nephrological system, and progressive pulmonary fibrosis [48].

Studies have been conducted on the impact of REMs on animal health but have been limited to a few REMs (Ce and La) and too short and medium-term observations (mostly 1–3 months) [1]. More long-term studies of exposure to REMs and lifelong observations are needed to fully understand the effects of REMs on animal health.

It is important to note that, due to the concerns about the spread and development of multidrug-resistant bacteria that are harmful to human health, the European Union banned the use of growth-promoting antibiotics in animal feed in 2006 [48]. Such restrictions led to the introduction of REMs as feed additives, which may have unintended consequences for animal health and welfare. Similar to the impacts of REMs on human health, animals also tend to develop respiratory, reproductive, and immune system dysfunctions [32]. Therefore, the use of REMs as feed additives should be carefully evaluated, and alternative strategies should be developed while also preventing the development of antibiotic resistance and food safety hazards.

## 8. REMs in Africa: The Imperative for Greater Research Coverage

While studies on REM exploitation are increasingly gaining attention globally, there has been far less research investigating the scramble for Africa's REMs (triggered by geopolitical concerns about China's near monopoly in the REM market) [50]. There is a great need to conduct REMs research on the scramble's implications on environmental and wider public health across the continent. Such studies are important and necessary given the history of less stringent safeguards in the continent's extractive industry, characterized by decades of significant adverse environmental health impacts.

An extensive review [50] of recent research in mining worldwide revealed that the criticality of REMs was one of the top five research trends. The authors found that, unsurprisingly, China, being the world's top producer of REMs, dominated the top-cited papers in this research stream. Other countries such as the United States, Australia, Russia, France, Estonia, and Asian countries such as India, Kazakhstan, and Malaysia were identified as existing or emerging sources of REM in the face of China's dominance [51]. References are also made to 'sizeable deposits' of REMs in Brazil, India, Australia, Canada, and Greenland [52]. However, these key articles, especially two of the three most cited [50,51], hardly recognized emerging trends in the exploration and exploitation of REMs in Africa and did not make references to the existing or emerging REM deposits in Africa. This is despite substantial evidence of mines such as Steenkampskraal in South Africa producing thorium since the 1950s/1960s [53], the US Geological Survey assessing promising REM sites in African countries such as Madagascar, Malawi, Mozambique, Tanzania, and South Africa as early as 2013 [53], with at least 11 advanced REE projects on the continent in 2015 [54].

In addition to these recognitions and coverage biases concerning the criticality of Africa's REM potentials in geopolitical debates about securing global REM supply chains, the authors argue that there are also significant implications and robust risk assessments

on the coverage of REM mining in Africa with a deep intellectual discourse; especially when calls for environmental concerns have been highlighted as a major consequence of the rapid advancement of REM exploitation outside China [51]. Below, the authors consider recent but relatively scarce studies not only highlighting substantial REM deposits or REM-related findings in key African countries, including Sierra Leone, South Africa, Cameroon, the Democratic Republic of Congo, and Nigeria, but also signposting the paucity of, and urgency for conducting studies on the environmental and health impacts of REM exploitation within Africa.

In Sierra Leonne, back in the 1920s, Akiwumi and D'Angelo note that the colonial government had discovered monatize, which contains REE cerium, neodymium, and lanthanum in some stream concentrates [55], and prospecting and exploitation occurred thereafter, then stopped around 1991 when the war began. Although foreign powers have been reported to have mined minerals to the economic disadvantage of Sierra Leone, environmental or health implications were not explored. Although exploitation has continued in recent times, illegal extraction, smuggling, and marginalization of local artisanal miners have also been reported. The negative environmental and health impacts of such informal and illegal mining activities are unfortunately of less attention in the media and academic papers within Africa.

In the Democratic Republic of Congo, already infamous for significant adverse environmental and public health impacts from its extractive industry, Kasay et al. [56] established evidence suggesting significant potential for REMs in the country, making it potentially *'A hub for future REE projects in Africa'*. However, very little emphasis on the associated environmental externalities of such REEs projects was made. In Cameroon, Temga et al. investigated the potential contamination of REEs in soils across the country by creating spatial distribution maps of REE concentrations [57]. Although no assessment of the potential environmental health risks of exploiting these REMs resources has been made, REMs mapping is still necessary to identify areas at risk of environmental impacts from REMs exploitation and could be an essential first step in establishing relevant future environmental policies and legislation on REMs mining.

Arguably, much more than any other African country, South Africa has been a major subject of REEs research, with investigations into the nature and behavior of REMs in soils and salt pan sediments in the country's Western Cape conducted as early as 2003 [58]. Jepson, [59] in a major 2012 paper, explored South Africa's two advanced REEs mines at the time, Steenkampskraal and Zandkopsdrift, which appeared to have positioned South Africa as the next frontier for REM production outside China. Jepson, cautious about invoking the terrible environmental precedence of REM mining in China on South Africa, concluded that the environmental effects of REM mining in the region are difficult to predict, rather choosing to focus on geopolitical implications, and perceptions of surrounding communities who seemed more concerned about expected positive socio-economic impacts on job creation than on environmental or public health impacts, the latter two which they seemed largely ignorant about, including risks associated with radioactive waste at the REM mines.

In addition to the potential impacts of REM exploitation in Africa, the rapidly growing influx of consumers of REM technology and electronic devices into the continent also portends significant environmental risks. Ayedun et al. (2015) revealed a substantial level of light REE enrichment, including lanthanum and cerium, in groundwater samples in the industrialized city of Lagos, Nigeria, attributed the identified REEs contamination to the ocean, manufacturing processes, and leaching from e-waste in the city [60]. Although not linked to the exploitation of REMs, these findings highlight the potential for adverse impacts resulting from exposure to REMs on drinking water quality and public health outcomes in industrialized parts of Africa. This calls for the need to consider clear and robust policy-making decisions on the disposal and management of REM-bearing waste in African countries already witnessing alarming rates of tech dumping from Europe and

America. Other studies [61,62] found correlations between higher levels of REEs in blood samples and hemoglobin deficiency in sub-Saharan Africans.

## 9. Planetary Health: A Holistic Approach for Addressing REMs Health Hazards

Planetary health refers to the highest attainable standards of human health along with the vitality of the natural systems that sustain life on Earth. This approach adopts a systems-based, solutions-oriented focus that works towards a more sustainable and equitable future for all [63,64]. In an era of green-tech revolution, the increasing demand for mining homegrown sources of REMs for reducing climate footprint has raised concerns about the cumulative radioactive impact of REMs on human, animal, and planetary health [4]. This requires proper regulations and responsible practices for mining and refining REMs. Moreover, these policymaking decisions should follow a sustainable and environmentally friendly protocol without displacement of indigenous communities, destruction of natural habitats, or causing adverse health conditions [65]. A holistic planetary health approach prioritizes long-term human, animal, and planetary health throughout the entire life cycle of REMs, from extraction to disposal. Promoting planetary health in relation to REMs is to focus on the social and economic dimensions of their extraction as well. This includes supporting local communities, reducing the negative social impact, improving working conditions, and promoting ecosystem health toward more sustainable and responsible production practices [66]. Planetary health also adopts a multidisciplinary approach that integrates circular economy, policy solutions, technological innovations, and community engagement [63]. Recognizing the importance of increasing public health awareness and education about the hazards associated with REMs. This includes e-waste, the radio-active fraction associated with the production of REMs, or the use of REMs as feed additives in livestock and as soil fertilizers, all of which can be considered anthropogenic drivers for the production and demand of global REMs [66]. By taking a planetary health-centered approach to REMs, we can ensure that these critical and vital high-tech minerals '*once marginalized in the chemistry circle*' are extracted and used in a way that supports both human health and the health of the entire ecosystems [63,64]. In other words, there is a need to be proactively aware that our technological advances do not come at the cost of human, animal, and environmental well-being [64]. In general, planetary health offers a holistic framework to address the complex and interconnected challenges posed by the mining and processing of REMs, while promoting sustainable development and safeguarding human and environmental health.

## 10. Recommendations

As REM exploitation and production continue to expand over the coming decades, more research and development within the African continent context warrants further elucidations. REM bioaccumulation has the potential to exceed toxicity thresholds, worsening the health of people, animals, the environment, and global health at large. Thus, the development of methods that would reliably estimate the concentrations and bioavailability of REMs, evaluate potential risks, assess ecosystem and health impacts, and propose technical solutions and policy pathways are all essentially needed as integrated actions since REM exploitation is already underway in some countries on the African continent. Moreso, future research should focus not only on providing didactic literature on proper waste disposal from REM processing, with details on suppression of dust particles from mines, and prevention of hazardous environmental impacts after mines have been closed, but also on identifying policies and regulatory approaches for ensuring eco-friendly and public health awareness of REM exploitation risks. This, along with the sustainable use and disposal of REM-bearing technological products, especially in African countries already experiencing a wave of e-waste dumping. Recommendations for using a planetary health approach to mitigate the hazards associated with REMs with a focus on the African context include:

1.  Promoting sustainable mining practices: encourage the adoption of sustainable practices that minimize environmental impact, such as reducing waste and conserving water and renewable energy sources. This can help reduce the environmental impact of REM extraction and processing in Africa.

2.  Increasing efficiency and recycling: Increase efficiency in the use of REMs by designing products that use fewer rare-earth materials or recycling existing resources. This can help reduce the demand for new REM extraction and processing.

3.  Encouraging research and innovation: explore alternative materials that can be used in place of REMs in high-tech products. For example, research is being conducted on using organic polymers instead of REMs in magnets [66]. This can help reduce the demand for new extraction and processing of REMs in Africa.

4.  Promoting ecosystem health: protect and promote ecosystem health by reducing pollution and preserving biodiversity. This can help to mitigate the health impacts of REMs on both humans and animals.

5.  Engaging local communities: Engage local communities and indigenous people in the decision-making processes related to the extraction and processing of REMs to ensure that their voices are heard and their rights are respected.

6.  Increasing transparency and accountability: Increase transparency in the REM supply chain to ensure that companies operate ethically and in accordance with environmental and social regulations. This should include ensuring that local communities are properly consulted and that their rights are respected.

7.  Conducting comprehensive environmental assessment: conduct a comprehensive environmental assessment before, during, and after REM mining and processing to identify and mitigate potential health and environmental risk. This should include an assessment of the potential impacts on local ecosystems and the health of nearby communities.

8.  Developing local capacity: support the development of local capacity for REM extraction and processing to create more jobs and economic opportunities for African communities. This can help to promote local development, regional, and inter-continental partnerships, and increase the value of raw materials produced in Africa.

9.  Enforcing environmental and social regulations: This ensures that companies operating in Africa are held accountable for their actions and that they operate ethically and in line with international standards.

10. Raising awareness: education and promoting a pro-planetary mindset are essential components to curb the negative impact of REM accumulation in the food chain towards food safety practices.

By implementing these recommendations, the hazards associated with REMs in Africa can be addressed towards a more sustainable and equitable future for humans, animals, and planetary health.

## 11. Conclusions

Transitioning to green technology and a climate-safe future requires a holistic planetary health lens that recognizes the multifaceted dimensions of REMs' impact at the human, animal, and ecosystem interface. Recycling and reusing REMs from waste materials or mine tailings is generally recognized to be a more environmentally friendly activity than the establishment of new REM mines. However, these recycling techniques may carry environmental hazards, such as high energy and chemical consumption. Globally, all types of mining have the potential to have negative consequences. However, this review has shown that the nature and scale of these impacts in Africa have been less clear, therefore necessitating an urgent need for more research to fully understand the effects of REMs on the continent. Proper mine management, adherence to adequate safety protocols, and sustainable processing methods, as well as e-waste handling systems, will go a long way toward ensuring that REM mining has fewer impacts on ecosystems and global health. Continued action with planetary health in mind should, therefore, be taken by mining

corporations and industrialists to manage REM exposure, particularly in Africa. This should span across different sectors, for example, non-governmental organizations to advocate and lobby for sustainable REM mining, processing, and waste disposal processes; and policymakers to formulate, execute, and enforce robust policies and regulations. All of these should be aimed towards reducing the risk of human, animal, and environmental exposure to REMs for generations to come.

**Author Contributions:** Conceptualization, H.J.O., Y.A.T., E.O.T. and A.O.M.; methodology, H.J.O., Y.A.T. and M.S.E.-S.; resources, H.J.O., Y.A.T., E.O.T., A.O.M., R.O.Y., I.O.O. and M.S.E.-S.; data curation, H.J.O., Y.A.T., E.O.T., A.O.M., R.O.Y. and I.O.O.; writing—original draft preparation, Y.A.T., H.J.O., E.O.T., A.O.M., R.O.Y., S.A.J., M.K.O., I.O.O., O.M.E., Y.I.S., A.F.A. and M.S.E.-S.; writing—review and editing, Y.A.T., H.J.O., E.O.T., I.O.O. and M.S.E.-S.; supervision, M.S.E.-S., Y.A.T., H.J.O. and I.O.O. All authors have read and agreed to the published version of the manuscript.

**Funding:** This research received no external funding.

**Institutional Review Board Statement:** Not applicable.

**Informed Consent Statement:** Not applicable.

**Data Availability Statement:** Not applicable.

**Acknowledgments:** The authors also appreciate the editor and reviewer(s) of the journal for their suggestions in improving the quality of the manuscript.

**Conflicts of Interest:** Mr. Emmanuel O. Taiwo has contributed to this work in a personal capacity, and the views expressed in this article do not represent the views of FCDO.

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
