# Peer review of "Global Environmental Health Impacts of Rare Earth Metals: Insights for Research and Policy Making in Africa"

_challenges, doi:10.3390/challe14020020_

Round 1
Reviewer 1 Report
Dear Authors,
The topic is current and interesting, but you have to support it with facts and analysis,
The introductory part is partially similar to your abstract, correct it
English must be rearranged
Why are there so many authors at work? what are the individual contributions of the author?
You did not indicate in the paper where the deposits of rare earths are in Africa and what their potential quantities are
In paragraphs 6.2 and 6.3 you should have listed some specific studies and results
Why didn't you mention in your paper some methods for solving the problem of the influence of REM on people, animals and the environment.
Author Response
Comments and Suggestions for Authors
Dear Authors,
The topic is current and interesting, but you have to support it with facts and analysis,
Dear Reviewer,
Thank you for taking the time to review our manuscript, your insightful suggestions and comments have, indeed, strengthen our manuscript. Here below, is how we have addressed your comments and suggestions point-by-point.
Comment:
The introductory part is partially similar to your abstract, correct it
Response:
This has been addressed in the revised manuscript.
Comment:
English must be rearranged
Response:
This has been addressed in the revised manuscript.
Comment:
Why are there so many authors at work? what are the individual contributions of the author?
Response:
This has been addressed in the revised manuscript. The Authors Contribution in the manuscript has been elucidated in the “Author Contribution Section”.
Comment:
You did not indicate in the paper where the deposits of rare earths are in Africa and what their potential quantities are
Response:
This has been addressed in the revised manuscript in lines 61-73.
Comment:
In paragraphs 6.2 and 6.3 you should have listed some specific studies and results
Response:
This has been addressed in the revised manuscript according to your suggestion.
Comment:
Why didn't you mention in your paper some methods for solving the problem of the influence of REM on people, animals and the environment.
Response:
A section on “Planetary health: A Holistic Approach for Addressing REMs Health Hazard” has been included in the revised manuscript. Also, the recommendation has been significantly improved to include suggestions for solving the problem of the influence of REM on people, animals and the environment.
We hope you are satisfied with our revisions. Thank you.
Sincerely,
Authors.

Reviewer 2 Report
This is an important and developing area and this is an admirable attempt to provide a broad overview of the background, future trends and risks to environment and human health from the extraction of Rare Earth Metals, with a particular focus on the implications for Africa.
I would have liked to see a bit more structure on how the literature was searched. I appreciate that this was not intended to be a systematic review but nevertheless it would be useful to know why this structure was chosen and how the literature was searched. For example, did this work arise from a series of workshops or a learning set? Did you sense check the format with academic peers, or stakeholders and, if the latter was this industry, regulators, NGOs, policy makers etc.?
Most importantly, did you use specific search terms for a literature search? For evidence of environmental impact did you look for environmental impact assessments or field sampling studies for REMs? For health impacts did you look for epidemiological research such as case-control studies or cohort studies and did you use specific search terms such as pneumoconiosis, interstitial lung disease, lung cancer, hepatic cancer, leukaemia, lymphoma or bone marrow disorder related to REM exposure?
There are several areas where statements are made but supporting references are lacking. For example, in the paragraph beginning on line 272, there are statements about the scramble for Africa’s REM resources and about less stringent safeguards in the continent’s extractive industries (both unreferenced). While both statements seem very plausible, it would have been useful to see references for supporting evidence, at least from news sources if not from peer reviewed papers.
Apart from these methodological issues, there are some parts of the text where the language could be clearer:
Lines 166-168 reads ‘However, several studies have demonstrated the beneficial effects of REMs in low concentrations and harmful effects in higher levels in terrestrial plants and also in the water among aquatic algae and plants’. Should this read but harmful effects at higher levels?
Line 181 refers to Indium as an REM but it was not mentioned in lines 50&51 ‘REMs are a group of seventeen metallic elements that comprise fifteen elements in the lanthanide series, in addition to scandium and yttrium’. If it is being included, and it is not in the lanthanide series, should it be added to scandium and yttrium in the introduction?
Lines 243&4 refer to lung health issues including lung cancer. This might read better as chronic lung disease (such as pneumoconiosis) and lung cancer.
Line 294 begins with Asides,.. The term Aside from… would be better English.
Author Response
Comments and Suggestions for Authors
This is an important and developing area and this is an admirable attempt to provide a broad overview of the background, future trends and risks to environment and human health from the extraction of Rare Earth Metals, with a particular focus on the implications for Africa.
Dear Reviewer,
Thank you for taking the time to review our manuscript, your insightful suggestions and comments have, indeed, strengthen our manuscript. Here below, is how we have addressed your comments and suggestions point-by-point.
Comment:
I would have liked to see a bit more structure on how the literature was searched. I appreciate that this was not intended to be a systematic review but nevertheless it would be useful to know why this structure was chosen and how the literature was searched. For example, did this work arise from a series of workshops or a learning set? Did you sense check the format with academic peers, or stakeholders and, if the latter was this industry, regulators, NGOs, policy makers etc.?
Response:
This has been addressed in the revised manuscript in lines 94-109.
Comment:
Most importantly, did you use specific search terms for a literature search? For evidence of environmental impact did you look for environmental impact assessments or field sampling studies for REMs? For health impacts did you look for epidemiological research such as case-control studies or cohort studies and did you use specific search terms such as pneumoconiosis, interstitial lung disease, lung cancer, hepatic cancer, leukaemia, lymphoma or bone marrow disorder related to REM exposure?
Response:
This has been addressed in the revised manuscript.
Comment:
There are several areas where statements are made but supporting references are lacking. For example, in the paragraph beginning on line 272, there are statements about the scramble for Africa’s REM resources and about less stringent safeguards in the continent’s extractive industries (both unreferenced). While both statements seem very plausible, it would have been useful to see references for supporting evidence, at least from news sources if not from peer reviewed papers.
Response:
This has been addressed in the revised manuscript in line 242.
Comment:
Apart from these methodological issues, there are some parts of the text where the language could be clearer:
Lines 166-168 reads ‘However, several studies have demonstrated the beneficial effects of REMs in low concentrations and harmful effects in higher levels in terrestrial plants and also in the water among aquatic algae and plants’. Should this read but harmful effects at higher levels?
Response:
This has been addressed in the revised manuscript in line 202.
Comment:
Line 181 refers to Indium as an REM but it was not mentioned in lines 50&51 ‘REMs are a group of seventeen metallic elements that comprise fifteen elements in the lanthanide series, in addition to scandium and yttrium’. If it is being included, and it is not in the lanthanide series, should it be added to scandium and yttrium in the introduction?
Response:
This has been addressed in the revised manuscript in line 52.
Comment:
Lines 243&4 refer to lung health issues including lung cancer. This might read better as chronic lung disease (such as pneumoconiosis) and lung cancer.
Response:
This has been addressed in the revised manuscript in line 295.
Comment:
Line 294 begins with Asides,.. The term Aside from… would be better English.
Response:
This has been addressed in the revised manuscript in line 362.
We hope you are satisfied with our revisions. Thank you.
Sincerely,
Authors.

Reviewer 3 Report
Dear authors,
Thank you for your interesting article which deals with important minerals that are gaining importance from a global perspective. After reading the article, I have assessed that the text presented does not have the basic attributes of an academic paper. The topic is very broadly conceived, the paper lacks methods and the conclusions of the paper do not bring new insights. Part of the thesis deals in detail with the historical aspects of some REMs. At the same time, there is a lack of wider literature. As far as the literature is concerned, especially environmental impacts, health aspects, etc., I can recommend the authors for further work, e.g. :
- Lindgato, 2022: Save the giants: demand beyond production capacity of tantalum raw materials
- Macháček, 2019: Typology of environmental impacts of artisanal and small-scale mining in African Great Lakes Region
- Elgstrand, K.; Sherson, D.L.; Jørs, E.; Nogueira, C.; Thomsen, J.F.; Fingerhut, M.; Burström, L.; Rintamäki, H. Safety and health in mining: Part 1. Occup. Health. S. Afr. 2017
- Macháček, 2020: Alluvial Artisanal and Small-Scale Mining in A River Stream—Rutsiro Case Study (Rwanda)
- Owusu et al, 2019: Small in size, but big in impact”: socio-environmental reforms for sustainable artisanal and small-scale mining
- Pedersen, A.F.; Nielsen, J.Ø.; Friis, C.; Jønsson, J.B. Mineral exhaustion and its livelihood implications for artisanal and small-scale miners. Environ. Sci. Policy 2021, 119, 34–43.
- Lindagato et al, 2023: Lithium Metal: The Key to Green Transportation
Continue the work, which is very interesting, but more academic approaches are needed.
Best regards
Author Response
Comments and Suggestions for Authors
Dear authors,
Thank you for your interesting article which deals with important minerals that are gaining importance from a global perspective. After reading the article, I have assessed that the text presented does not have the basic attributes of an academic paper. The topic is very broadly conceived, the paper lacks methods and the conclusions of the paper do not bring new insights. Part of the thesis deals in detail with the historical aspects of some REMs. At the same time, there is a lack of wider literature. As far as the literature is concerned, especially environmental impacts, health aspects, etc., I can recommend the authors for further work, e.g.:
- Lindgato, 2022: Save the giants: demand beyond production capacity of tantalum raw materials
- Macháček, 2019: Typology of environmental impacts of artisanal and small-scale mining in African Great Lakes Region
- Elgstrand, K.; Sherson, D.L.; Jørs, E.; Nogueira, C.; Thomsen, J.F.; Fingerhut, M.; Burström, L.; Rintamäki, H. Safety and health in mining: Part 1. Occup. Health. S. Afr. 2017
- Macháček, 2020: Alluvial Artisanal and Small-Scale Mining in A River Stream—Rutsiro Case Study (Rwanda)
- Owusu et al, 2019: Small in size, but big in impact”: socio-environmental reforms for sustainable artisanal and small-scale mining
- Pedersen, A.F.; Nielsen, J.Ø.; Friis, C.; Jønsson, J.B. Mineral exhaustion and its livelihood implications for artisanal and small-scale miners. Environ. Sci. Policy 2021, 119, 34–43.
- Lindagato et al, 2023: Lithium Metal: The Key to Green Transportation
Continue the work, which is very interesting, but more academic approaches are needed.
Response,
Dear Reviewer,
Thank you for taking the time to review our manuscript, your insightful suggestions and comments have, indeed, strengthen our manuscript. In the revised manuscript, the method has been included and the recommendation and conclusion of the paper has been improved to bring new insights. The literature on REMs has been improved with the papers recommended to provide more details on environmental impacts and health aspects of REMs. Also, a section on “Planetary health: A Holistic Approach for Addressing REMs Health Hazard” has been included Overall, the manuscript has been significantly improved.
We hope you are satisfied with our revisions. Thank you.
Sincerely,
Authors.

Round 2
Reviewer 1 Report
Dear authors,
I am generally satisfied with the way you responded to the remarks and suggestions, my opinion is that your work should be accepted in the journal.
Author Response
Comments and Suggestions for Authors
Dear Authors,
I am generally satisfied with the way you responded to the remarks and suggestions, my opinions are that your work should be accepted in the journal.
Dear Reviewer,
Thank you for taking the time to review our manuscript, your insightful suggestions and comments have, indeed, strengthen our manuscript. Thank you for recommending our work for acceptance.
Sincerely,
Authors.
Reviewer 2 Report
The paper is much improved.
It is now clear that this is a narrative review, using a selection of key phrases, which uncovered 66 articles. However, the search methodology does not allow many conclusions about the nature or quality of the articles that are included. The section on planetary health adds an important perspective.
There are still places where the English and terminology could be improved. For example, in the abstract, I am not sure that 'limited studies are exploring' is an improvement on 'there are limited studies exploring' (line 33) and 'therefore calling for more research' might read better as 'meaning that more research is needed (lines 38 and 39).
There also seems to be inconsistent usage of the plural for the terms REMs / REEs; I wondered if at some point a 'replace all' instruction had been used to make the terms plural but, for examle, it would be better to use REM deposits and REM reservoirs and REM repositories (NOT respositories).
Also there are sections (see Section 9 on Planetary Health where REMs and REEs seem to be used interchangeably? As the Introduction states that REMs are also known as REES, it might have been better to use REM more consistently unless quoting directly from a source that uses REE (e.g. LREEs and HREES).
If there are fifteen elements in the lanthanide series, in addition to scandium, indium and yttrium being included, there are several places where the paper should refer to eighteen REMs – check on Page 2 (Line 51) and Page 4 (line 159).
Author Response
Comments and Suggestions for Authors
Dear Reviewer,
Thank you for taking the time to review our manuscript, your insightful suggestions and comments have, indeed, strengthen our manuscript. Here below, is how we have addressed your comments and suggestions point-by-point.
Comment:
The paper is much improved.
It is now clear that this is a narrative review, using a selection of key phrases, which uncovered 66 articles. However, the search methodology does not allow many conclusions about the nature or quality of the articles that are included. The section on planetary health adds an important perspective.
Response:
The methodology section has been improved to allow many conclusions about the nature of the articles used.
Comment:
There are still places where the English and terminology could be improved. For example, in the abstract, I am not sure that 'limited studies are exploring' is an improvement on 'there are limited studies exploring' (line 33), and 'therefore calling for more research' might read better as 'meaning that more research is needed (lines 38 and 39).
Response:
This has been addressed in the revised manuscript.
Comment:
There also seems to be inconsistent usage of the plural for the terms REMs / REEs; I wondered if at some point a 'replace all' instruction had been used to make the terms plural but, for example, it would be better to use REM deposits and REM reservoirs and REM repositories (NOT respositories).
Response:
This has been addressed in the revised manuscript.
Comment:
Also, there are sections (see Section 9 on Planetary Health where REMs and REEs seem to be used interchangeably? As the Introduction states that REMs are also known as REES, it might have been better to use REM more consistently unless quoting directly from a source that uses REE (e.g. LREEs and HREES).
Response:
This has been addressed in the revised manuscript.
Comment:
If there are fifteen elements in the lanthanide series, in addition to scandium, indium and yttrium being included, there are several places where the paper should refer to eighteen REMs – check on Page 2 (Line 51) and Page 4 (line 159).
Response:
This has been addressed in the revised manuscript.
We hope you are satisfied with our revisions. Thank you.
Sincerely,
Authors.
Reviewer 3 Report
Thank you for the added text.
Author Response
Comments and Suggestions for Authors
Thank you for the added text.
Dear Reviewer,
Thank you for taking the time to review our manuscript, your insightful suggestions and comments have, indeed, strengthen our manuscript.
Sincerely,
Authors.